# Adherence to Cervical Cancer Screening Programs in Migrant Populations: A Systematic Review and Meta-Analysis

**DOI:** 10.3390/ijerph20032200

**Published:** 2023-01-26

**Authors:** Isabella Rosato, Teresa Dalla Zuanna, Valentina Tricarico, Claudio Barbiellini Amidei, Cristina Canova

**Affiliations:** 1Unit of Biostatistics, Epidemiology and Public Health, Department of Cardio-Thoraco-Vascular Sciences and Public Health, University of Padua, 35131 Padua, Italy; 2Epidemiological Department, Azienda Zero, Veneto Region, 35131 Padua, Italy

**Keywords:** cervical cancer screening, adherence rate, migrants, HMPC countries

## Abstract

Organized cervical cancer screening programs to promote the early identification of precancerous lesions have proven to be effective in decreasing the burden associated with cervical cancer, but knowledge regarding screening adherence among migrant women compared to that of native women has not been summarized. A systematic search of the literature on PubMed, Scopus and Embase led to the identification of 772 papers that were published up to July 2022 and reported population-based data regarding adherence to cervical screening. The screening participation rates among migrant women, compared to native women, were pooled using a random-effects meta-analysis. A total of 18 papers were included in the review, with most of them being conducted in Europe (83.3%). Overall, migrants showed a significantly lower participation rate compared to native women (OR for screening adherence: 0.54, 95% CI = 0.42–0.70). This discrepancy was especially evident for migrant women from North Africa and Sub-Saharan Africa (OR = 0.47, 95% CI = 0.35–0.63, and OR = 0.35, 95% CI = 0.24–0.49, respectively). The results of this systematic review emphasize the importance of increasing cervical cancer screening adherence among migrant women. A significant heterogeneity in screening adherence was observed based on the country of origin. Interventions aimed at reducing the disparities in screening participation should specifically consider how to improve the recruitment of migrant women.

## 1. Introduction

Cervical cancer is currently listed as one of the most commonly diagnosed cancers in women worldwide [1]. Numerous types of human papilloma viruses (HPVs) are responsible for common infections in women that may persist or regress spontaneously. Cervical cancer development is linked to persistent infection with the oncogenic types of HPV [2,3]. Individuals are often unaware of the ongoing infection, and precancerous lesions are mostly asymptomatic [4].

Primary prevention strategies include implementing safe sex habits and HPV vaccination; bi-, quadri- and nine-valent HPV vaccines have been progressively developed starting from 2006 [4]. Secondary prevention strategies are focused on the early detection of subclinical forms of pre-cancerous or cancerous lesions [5,6]. Cervical cancer screening has traditionally been based on cytology (Papanicolaou test, also known as Pap smear or smear test). In the presence of positive cytological results, the diagnosis is confirmed by a colposcopy, and appropriate treatment is informed by a biopsy of suspicious lesions for histological diagnosis. Newer screening tests introduced in the last 15 years include visual inspection with acetic acid (VIA) and molecular tests, mainly high-risk HPV DNA-based tests, which are suitable for use in all settings [7]. Screening performed using HPV DNA-based testing ensures a standard screening quality, while showing an increased sensitivity and accuracy compared to cervical cytology, with its high negative predictive value allowing a 5-year screening interval, which is longer than the 3-year interval required for Pap smear [8,9,10]. Following the evidence that underlines that HPV DNA-based testing is more accurate and has better reproducibility than Pap test, most recent European guidelines now recommend using HPV DNA-based testing as a primary screening strategy [10]. Despite the presence of effective screening strategies and the WHO’s recommendations to prioritize cervical cancer screening in women aged 30–49 years [7], two in three women in this age range have never been screened for cervical cancer [11]. Moreover, there are large differences in screening participation rate across different global regions. The rollout of screening is very low in low-income and middle-income countries, where the burden associated with the disease is the highest [11].

The implementation of organized cervical screening programs has proven to be effective in decreasing the burden of the disease and subsequent mortality [12]. Most European countries offer organized free-of-charge screening, whereas in settings in which screening is opportunistic, payment and reimbursement may depend on women’s healthcare insurance [13]. Organized screening programs rely on specific defining elements: a well-identified target population; the implementation of a population-based register; and the existence of quality control procedures, screening pathways and epidemiological monitoring of the effectiveness of the program itself [14]. Organized cervical screening programs are more efficient than opportunistic approaches, and they provide wider coverage by ensuring that invitations to participate reach the target population [9,10]. In countries with organized cervical screening programs, knowledge about the disparities in cervical screening participation among migrants compared with natives is sparse and not summarized yet [15]. Research on this topic is often limited due to missing information regarding the country of origin, as well as cultural and linguistic barriers. When studies rely on surveys to collect information regarding screening participation, a problem is posed by the limited number of subjects investigated and the presence of recall bias [16]. Participation in organized screening programs is offered to both native and migrant women without differences, but many studies have shown that, in various settings, foreign-born women are less likely to take part in cervical screening compared to native women, with subsequent increased risk for late diagnosis of the disease and reduced screening effectiveness [15,16,17].

Our main objective is to summarize current evidence regarding organized cervical screening attendance among migrant women and to compare their participation with that of native women. We also aim to identify migrant subgroups that are characterized by the lowest attendance.

## 2. Materials and Methods

### 2.1. Literature Search

A systematic review was conducted following the PRISMA Guidelines for systematic reviews and meta-analyses [18]. PubMed, Scopus and Embase databases were searched from inception up to July 2022. Following the PICO (Population, Intervention, Comparison, Outcome) framework, we intended to evaluate the attendance rate (O) of migrant women with the correct age range for participation (P) in cervical cancer screening programs (I), compared to that of native women with the same characteristics (C), using population-based studies. For each database, the search was conducted using a combination of subject headings and free text words for the population (migrant women) and the outcome considered (cervical cancer screening). Appendix A outlines the search strategy adopted for PubMed, Scopus and Embase databases.

For the purposes of our review, we defined “migrants” as women born in countries different from the one in which the study was conducted, as defined in a review conducted on similar topics [19], as well as women with a different citizenship or mother tongue.

Studies that investigated only refugees and asylum seekers’ participation in cervical screening were excluded from the present review because their characteristics differ greatly from those of other groups of migrants. Subjects who migrate on the basis of socio-economic reasons make a conscious, voluntary choice to leave their country of origin and can eventually return home in safety, if desired [20,21]. On the other hand, refugees and asylum seekers undergo forced migration; during their early resettlement, they experiment unique challenges associated with health care utilization and access, and women forming part of these subgroups show different priority health issues compared to other migrant women [22,23,24].

We excluded from the review studies that presented only self-reported information regarding cervical screening participation obtained through surveys and questionnaires, as self-reporting shows validity issues in determining individuals’ screening history and can lead to an underestimation or an overestimation of screening prevalence, especially in socially disadvantaged subgroups [25]. We also excluded studies that reported participation in cervical screening through self-sampling strategies because these are not included nowadays in all organized screening programs [26]. Moreover, in most of the published studies, self-sampling approaches have been proposed as an alternative screening method for specific subgroups of non-attendees or women who actively decline traditional screening [27,28].

For studies that investigated the same populations, we kept the most recent one or the one with the highest number of participants. The absence of data regarding the native group’s participation in the screening (control group) was considered as an exclusion criterion. Only papers published in English were included. The inclusion and exclusion criteria are fully specified below (Table 1).

This research was conducted in three steps. In the first step, we identified and removed duplicates. In the second step, two reviewers (IR and VT) screened all identified papers independently by title and abstract, selecting only those that focused on cervical cancer screening among migrant women. The third step consisted of an independent screening of the selected papers by full text. At each stage, disagreements between the reviewers were solved by the intervention of a third reviewer (TDZ). After the second stage, all references of the selected studies were also checked for any additional relevant papers. The screening process was conducted using the Covidence software (Covidence systematic review software, Veritas Health Innovation, Melbourne, Australia. Available at www.covidence.org, accessed on 6 October 2022).

### 2.2. Data Extraction and Statistical Analysis

For each selected paper, the following information were extracted: title, authors, publication year, country, characteristics of the screening program in the selected country, screening compliance definition according to the specific study, characteristics of the study population (inclusion and exclusion criteria, total number of migrants and natives participating in the study, and classification and stratification strategies used to define migrants), rates of participation in cancer screening for natives and migrants (numerator and denominator), migrant categories based on the country of origin, and unadjusted and adjusted odds ratios (ORs) for screening participation for the migrant populations. When the ORs were not readily available, we calculated unadjusted ORs (with relative standard error) for cervical screening participation among migrant women, compared to native women.

To account for the variability in settings and populations investigated in the included studies, we chose a random-effects model for the meta-analysis and pooled unadjusted ORs for screening participation in a forest plot. The heterogeneity among the studies was investigated using I^2^ statistic and was defined as low, moderate or high using the I^2^ cut-offs of 25%, 50% and 75%, respectively [29]. Stratified meta-analyses were conducted, with forest plots presenting the information on specific migrant subgroups coming from high migratory pressure countries (HMPCs): Central-Eastern (CE) Europe, Sub-Saharan (SS) Africa, North Africa, Asia and Central-South (CS) America. Stratified analyses for migrants coming from HMPCs were chosen because their socioeconomic characteristics and health habits generally differ greatly from those of migrants coming from low migratory pressure countries (LMPCs) and native subjects [30]. Given the high number of studies conducted in Northern Europe only (Denmark, Sweden, Norway and Finland), this subgroup of studies was also analyzed separately from the others. In the sensitivity analyses, we excluded studies that could be considered as potential outliers after an examination of the studentized residuals.

Meta-regression models and stratified analyses were conducted in an attempt to address the heterogeneity of the included studies. Publication bias was evaluated by checking for the presence of asymmetry in the funnel plot and using the regression test. A *p*-value < 0.05 was considered statistically significant. Statistical analyses were conducted using R (R Core Team (2022). R is a language and environment for statistical computing. (R Foundation for Statistical Computing, Vienna, Austria. URL https://www.R-project.org/, accessed on 1 December 2022).

## 3. Results

A total of 772 papers were retrieved using the search string. Figure 1 shows the study selection process in detail.

Full-text screening led to the exclusion of two studies [31,32] conducted on the same populations, which were investigated in more recent publications that were included in the review. A total of 18 papers were selected for the review. The main characteristics of the studies and the screening programs are summarized in Table 2 (more information in Appendix A).

Most of the included studies were conducted in Europe (n = 15, 83.3%), while the remaining were conducted in Australia (n = 2, 11.1%) and Canada (n = 1, 5.6%). The migrant populations were classified according to the country of birth/origin (n = 13, 73.7%), mother tongue (n = 2, 10.5%), citizenship (n = 1, 5.3%), or using more than one of these definitions (n = 2, 10.5%). The mean number of migrant women included in the included studies was 126,159 (median = 50,250, range: 1790–500,381), and the mean number of native women included was 678,037 (median = 304,773, range: 789–4,017,764) (Table 3). The mean participation in cervical cancer screening was considerably lower for migrant populations (19.1% versus 62.3% in natives) (Table 3).

Overall, migrant women had a 46% lower chance of participating in cervical cancer screening (OR: 0.54, 95% CI = 0.42–0.70, I^2^ = 100%) compared to native women (Figure 2).

Differences arose when analyses were conducted stratifying migrants according to the macro-areas of origin. The lowest OR for screening participation was observed for women coming from Sub-Saharan Africa (OR = 0.35, 95% CI = 0.24–0.49, I^2^ = 99%), followed by migrant women from North Africa (OR = 0.47, 95% CI = 0.35–0.63, I^2^= 97%). Women from Asia showed an OR for screening participation of 0.54 (95%CI = 0.40–0.73, I^2^= 100%), and those from Central-Eastern Europe had an OR of 0.58 (95% CI = 0.46–0.73, I^2^= 100%). Finally, women from Central-South America had the highest OR for screening participation, corresponding to 0.69 (95% CI = 0.58–0.82, I^2^ = 99%) (Figure 3). When analyzing only studies conducted in Northern Europe (Norway, Finland, Denmark, and Sweden) and when conducting the sensitivity analyses, no reduction in the amount of heterogeneity could be observed (Appendix A). The meta-regression models conducted using the participants’ mean age, the definition of migrant subjects (country of birth/origin, citizenship, mother tongue or more than one of the previous definitions), the type of test used (Pap test only or Pap test and HPV DNA test), or the publication year as covariates did not show a significant reduction in the observed heterogeneity. The funnel plot examination and regression tests did not show asymmetry (Appendix A).

## 4. Discussion

Overall, the results confirm that migrant women, irrespective of their country of origin, have a significantly lower adherence to cervical screening compared to native women; screening participation is extremely low for women coming from North and Sub-Saharan Africa and higher for women coming from Central and South America.

The disparities in screening participation could be explained by the presence of several aspects that influence adherence [42]. Studies focused on screening adherence have identified the most common barriers experienced by migrants, including economic, cultural, language, healthcare system-related, knowledge-related and individual-level barriers [49]. Socio-economic status (which compasses income, education and occupational class) could play a relevant role in the lower adherence among migrant groups, as it is known that the health condition of migrants, in comparison to natives, progressively converges toward the health behaviors and epidemiological profiles of the lowest socio-economic groups of the host populations [50]. People with lower socio-economic status show a lower adherence to preventive measures, such as cervical cancer screening [51,52,53]. In countries where screening is opportunistic, financial availability plays a significant role in participation [54]. Organized screening programs are, instead, free of charge or require the payment of a small contribution, and screening adherence should not be affected by income [55]. Although it is true that different economic resources might give access to more timely private services, it is more likely that factors other than income play a major role in determining the disparities in the uptake of cervical cancer screening programs. In fact, a study conducted in England found that ethnicity and education were the most important predictors of adherence to screening, while other indicators of wealth did not significantly affect adherence [56,57].

Among other relevant barriers to cervical screening adherence are the lack of information on screening opportunities and on its importance, the lack of appropriate language skills, and the presence of fear and discomfort related to screening procedures [56,58].

Language and cultural barriers undermine both the access and the quality of health services for migrants [59]. Providing information on cervical cancer screening in several languages and adapting the information for different cultural backgrounds are the means to tailoring screening programs at a relatively low cost. Impersonal communication through printed materials may not work, and community networks may be the most effective way to reach migrants [60].

Different beliefs can also influence participation in screening procedures. Examples of cultural attitudes and beliefs include fatalism, lack of perceived vulnerability, and unfamiliarity with the concept of screening [61], as well as the common thought that the procedure itself may play a role in the risk of cancer development [62].

Among other possible solutions that can improve the adherence of migrants and other deprived groups is the adaptation/modification of screening program organization. The latest WHO recommendations suggest using HPV DNA detection as the primary screening test [7], and it has already been introduced in some countries [35]. This test could increase the compliance of some hard-to-reach populations by reducing the frequency of screening and by allowing self-sampling with less invasive procedures [63]. Although it is not sure whether this new screening tool will reduce existing inequalities, it will probably increase population coverage [7,45].

The studies included in this review reported different levels of attending universal screening programs in the reference group of native women, and there was great variability in the number of participants and in the settings, as well as in the classification of countries used to identify the migrants’ origin.

For all the subgroups investigated, participation is lower compared to that of native women, but the discrepancy is particularly evident for subgroups originating from Sub-Saharan Africa and North Africa, followed by migrants from Asia. Migrants coming from Africa, especially from Sub-Saharan African countries, to higher income countries are found to have the worst health outcomes when compared to natives, for almost any health indicators considered [63,64,65]. Health practices based on the traditional medicine and cultural background of migrants from Africa and Asia may be dissimilar to those of European people and health professionals, and it is essential to take particular care when dealing with these groups [66]. For some migrant groups, a Pap smear may represent an invasive and personal procedure or could be associated with the stigma following oncological diseases [67], thus posing cultural barriers and hindering these women from utilizing screening services [68,69]. In this sense, it has been shown that having a family doctor who is from the same country of origin as the women significantly increases the chance of being screened [64] and that the gender of healthcare professionals impacts the level of screening participation [70]. The evidence suggests that interpreter services, often provided by cultural mediators, as well as the promotion of diversity among health professionals by recruiting staff with varied linguistic and cultural skill, are effective strategies to overcome or at least reduce the disparities in screening participation [50].

Overall, our results show that relevant differences in cervical cancer screening adherence exist among migrant women from different geographical macro-areas. Migrants coming from low migratory pressure countries (LMPCs) were not considered in the stratified analyses because previously published studies show that they are generally characterized by a similar health profile to that of the native populations [71]. These findings underline that migrant women should not be considered as a homogenous group and specific tailored strategies should be implemented to support specific subgroups. The significantly low screening adherence emphasizes the importance for healthcare services to find ways to better engage with migrant groups to address their health needs more appropriately. This includes tackling the barriers that prevent migrant women from accessing preventive care, such as cervical cancer screening programs [50]. It is also essential to train healthcare professionals to increase cultural competence to improve understanding regarding cultural diversity among patients [42].

To our knowledge, this is the first review that systematically summarizes the differences in rates of cervical cancer screening adherence among migrant and native women. The specificity of the inclusion criteria, together with the elevated number of subjects included in the meta-analysis, suggests that our results are robust. Our review is also characterized by some limitations. First, the inclusion of papers written only in English might have led to missing information regarding participation in screening among migrants reported in other languages. Only partial information regarding the migrants’ countries of origin was retrieved from the selected papers as several different classification methods were used to group the migrants. Moreover, the stratified analyses and the meta-regression models that were conducted considering the extracted principal variables did not lead to any reduction the observed heterogeneity, and it was not possible to identify other possible sources that were responsible of the high heterogeneity that was present among our included studies. The generalization of the results obtained from the analyses stratified by the country of origin needs to be interpreted with caution, since certain groups comprise large geographical areas and populations with different health beliefs, such as the group from Asia.

## 5. Conclusions

In countries with organized cervical screening programs, greater attention needs to be given to migrant women, who have lower participation rates compared to those of native women. A significant variation in screening adherence is observed based on the country of origin. Tailored strategies need to be implemented to adequately address migrants’ needs to increase screening adherence. These populations are at higher risk for late diagnosis of cervical cancer, with subsequent increased mortality risks and poorer health outcomes.

## Figures and Tables

**Figure 1 ijerph-20-02200-f001:**
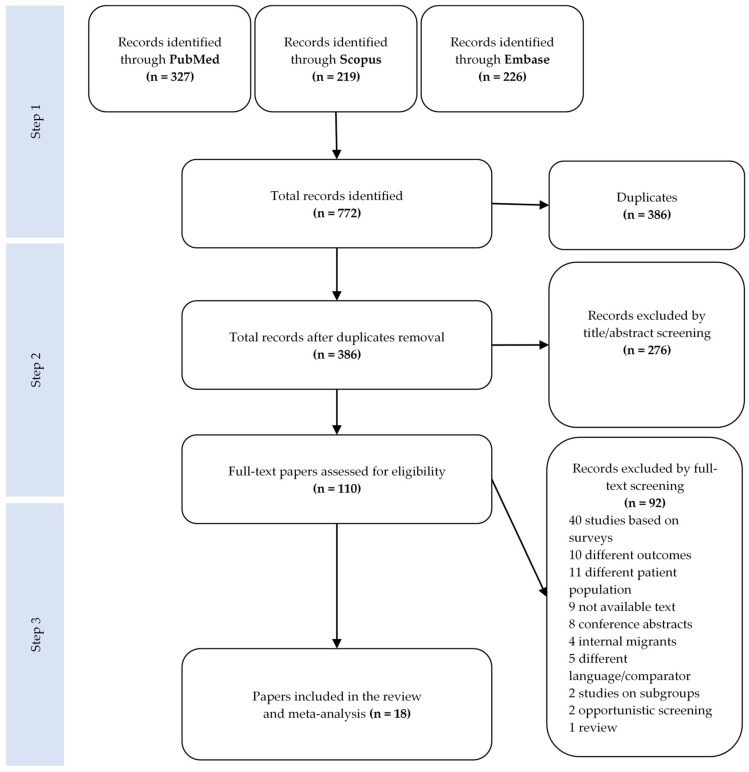
PRISMA flowchart for the paper selection process.

**Figure 2 ijerph-20-02200-f002:**
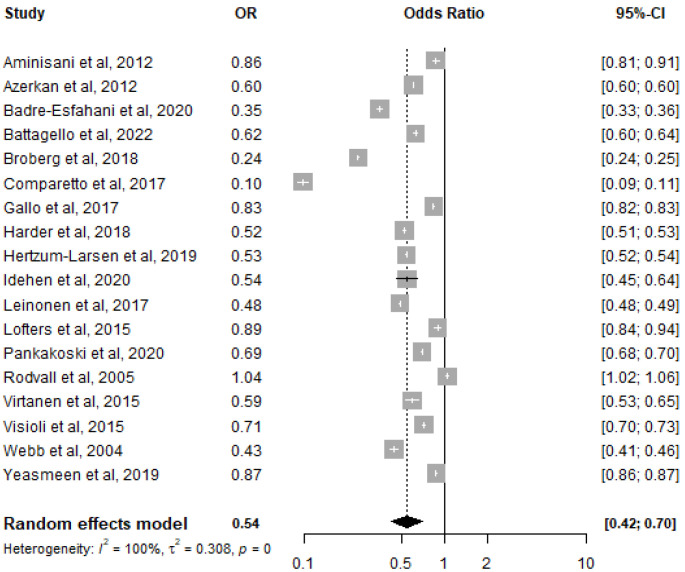
Forest plot for screening participation among all migrant populations, compared to native populations [16,25,33,34,35,36,37,38,39,40,41,42,43,44,45,46,47,48].

**Figure 3 ijerph-20-02200-f003:**
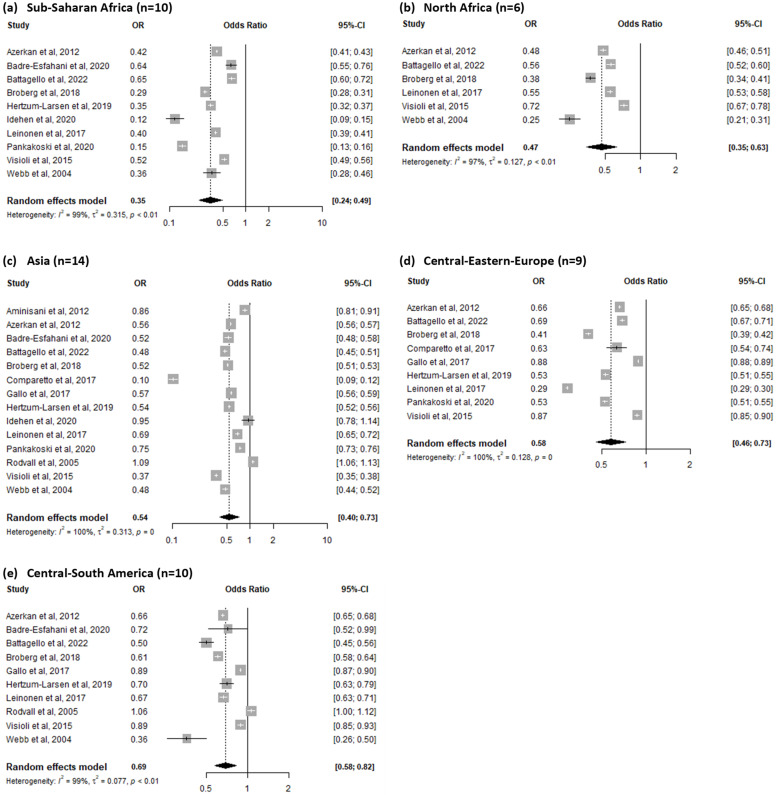
Forest plot for screening participation among Sub-Saharan African (**a**), North African (**b**), Asian (**c**), Central-Eastern European (**d**) and Central-South American (**e**) migrants compared to native populations [16,33,34,35,36,37,38,40,41,42,43,44,46,47].

**Table 1 ijerph-20-02200-t001:** Inclusion and exclusion criteria.

Inclusion Criteria	Exclusion Criteria
Original papers published up to July 2022 English language	Reviews, conference abstracts, commentaries, editorials, letters to Editor, and pilot studies Languages different from English
**Subjects**: female migrants aged 18+ and native controls (age range for screening invitation may vary in different countries)	**Subjects**: selected groups of participants (patients at high risk of cervical cancer, asylum seekers/refugees and internal migrants)
**Outcome**: measure of participation in organized cervical cancer screening (both Pap smear and HPV DNA test are included) **Data source**: population-based data (national registries and databases)	**Outcome**: participation in cervical cancer screening through self-sampling strategies or opportunistic screening **Data source**: surveys and questionnaires (self-reported measures)

**Table 2 ijerph-20-02200-t002:** Characteristics of the included studies.

Reference	Location	Screening Program Characteristics	Data Sources	Definition of Adherent Subject	Exclusion Criteria	Definition of Migrants	Characteristics of the Sample
Aminisani et al., 2012 [16]	Australia	Age: 18–69 years Time interval: 2 years	NSW Pap Test Register, NSW Midwives Data Collection	Pap test register screening record in the calendar/fiscal year 2001–2002, and screening performed within 2–3 years of giving birth	Women died after giving birth/during the follow-up period, abnormal cervical test, and unsatisfactory result in the 5 years preceding the study period	Country of birth	Migrants: women aged 20–54 years giving birth between 1 January and 31 December 2000; natives: women matched by birth in the same period, 5-year age group and residence area
Azerkan et al., 2012 [33]	Sweden	Age: 23–50 years Time interval: 3 years Age: 51–60 Time interval: 5 years	National Cancer Screening Register, and Swedish Total Population Register	Women aged 23–50 years were considered participants for 3 years from the last test, and women aged 51–60 years for 5 years after the last test	Missing information, emigration out of Sweden, death, and carcinoma in situ/invasive cancer before entry to the cohort	Country of birth	Migrants and native women with data from 1993 to 2005
Badre-Esfahani et al., 2020 [34]	Denmark	Age: 23–49 years Time interval: 3 years Age: 50–64 years Time interval: 5 years	Danish Civil Registration System, and Danish Pathology Register	Women with at least one registered cytology sample between the age of 22.5 and 24 years	Women with cervical cytology obtained before the age of 22.5 years, history of surgical removal of cervix, and diagnosis of cervical cancer	Country of origin	Migrants: women born during the period of 1985–1993 outside Denmark or with two immigrant parents. Natives: women born in Denmark in the same period
Battagello et al., 2022 [35]	Italy	Age: 25–29 years Time interval: 3 years Age: 30–64 years Time interval: 5 years	Cervical cancer screening databases from Local Health Units	Women screened after invitation	Women with spontaneous HPV vaccination and Western country citizenship, and undelivered invitations	Country of origin/ citizenship	Foreign and Italian women residing in the study area born between 1986 and 1992 and invited for the 1st time between 2011 and 2017
Broberg et al., 2018 [36]	Sweden	Age: 23–50 years Time interval: 3 years Age: 51–60 Time interval: 5 years	Swedish Total Population Register, Swedish National Cervical Screening Registry, and Statistics Sweden	Women who attended a screening program within 90 days of invitation between 1 January 2012 and 31 December 2012	Women migrated during the study period, with total hysterectomy, without regular invitation in 2012, and with regular invitation in 2012 but not participating within 90 days	Country of birth	Immigrant and Swedish women between 30–60 years of age on 31 December 2012
Comparetto et al., 2017 [37]	Italy	Age: 25–64 years Time interval: 3 years	Archives of LHU Serviceable Registry of Prato Province, and Cancer Registry of Tuscany	Woman respondents to the invitation	Women who did not receive the invitation letter, underwent a hysterectomy since the last test, and had a cervical test in the 12 months before the invitation	Citizenship	Residents of Prato with at least 1 invitation between 1 July 2004 and 30 June 2007
Gallo et al., 2017 [38]	Italy	Age: 25–64 years Time interval: 3 years	Screening program archives	Attendees at first appointment or at recall 1 month later	Not available	Country of birth/ citizenship	Women resident in Piedmont who received at least one invitation during the period of 2001–2013
Harder et al., 2018 [39]	Denmark	Age: 23–49 years Time interval: 3 years Age: 50–64 years Time interval: 5 years	Pathology Databank, National Patient Register, Cancer Register, Medical Birth Register, Prescription and Psychiatric Research Register, and Statistics Denmark	Women with a cervical cytology registered within the 4-year follow-up period	Women with missing information, with hysterectomy registered before baseline or in follow-up, and emigrated or died during follow-up	Country of origin	Migrant and native women invited for routine cervical cancer screening in 2008–2009
Hertzum-Larsen et al., 2019 [40]	Denmark	Age: 23–49 years Time interval: 3 years Age: 50–64 years Time interval: 5 years	Civil Registration System, Pathology Databank, National Patient Register, National Health Service Register, Medical Birth Register, Psychiatric Central Register, and Employment Register	Cervical cytology registered in the Pathology Databank during follow-up	Women born in Denmark to immigrant parents, were not residing in Denmark, were unsubscribed or fully hysterectomized, were pregnant, had missing data, and had not resided continuously in Denmark during the study period	Country of origin	Migrant and native women invited to the screening program between 2008 and 2009
Idehen et al., 2020 [41]	Finland	Age: 30–60 years Time interval: 5 years	Finnish National Population Registry, Mass Screening Registry, Care Register, Medical Birth Register, Register of Induced Abortions, Statistics Finland, and Social Insurance Institution of Finland	Positive response to invitation in 2008–2012	Males, women aged < 30 years, and not invited women	Country of origin	Migrant and native women invited to the organized cervical screening program during the period of 2008–2012
Leinonen et al., 2017 [42]	Norway	Age: 25–69 years Time interval: 3 years	Norwegian cervical cancer screening program system	Screening test recorded in 2008–2012	Incomplete screening history, women with missing data, women who had opted out from the program, women with previous diagnosis of gynecological cancer, and women under surveillance for cervical abnormalities	Country of origin	Migrant and native women alive and were a resident in Norway on 31 December 2012
Lofters et al., 2015 [25]	Canada	Age: 25–69 years Time interval: 3 years	Cytobase (Pap test registry)	Record of screening in the 3 years before study period	Women not eligible for health insurance coverage, hysterectomized, and with previous diagnosis of cervical cancer	Country of birth	Migrant and native women eligible for cervical screening from 2000 to 2007
Pankakoski et al., 2020 [43]	Finland	Age: 30–60 years (up to 25–65)	Mass Screening Registry, Population Registry, and Statistics Finland	Women who attended organized screening in the 5-year interval studied	Women with no information on socioeconomic status, mother tongue or home municipalities	Mother tongue	Migrant and native women born in 1950–1984 and residing in Finland in 2010–2014
Rodvall et al., 2005 [44]	Sweden	Age: 25–40 years Time interval: 3 years Age: 41–59 years Time interval: 4 years	Dataset of invited women, and National Longitudinal Population Database	Women having taken a smear within the program one year of receiving the invitation	Women not invited due to a recent smear, including those taken as an opportunistic screening test	Country of birth	Migrant and native women invited to the screening program between 1994 and 1996
Virtanen et al., 2015 [45]	Finland	Age: 30–60 years (up to 25–65)	Mass Screening Registry, and Statistics Finland	Women who attended screening after receiving an invitation letter in 2011–2012	Not invited women, women with missing information, emigrated, dead, and moved to other municipalities during the study	Mother tongue	Migrant and native women invited to screening between 2011 and 2012
Visioli et al., 2015 [46]	Italy	Age: 25–69 years Time interval: 3 years	Archive of invitations to the screening and archive of the Research Institute laboratory	Pap test performed within one year from the date of invitation	Missing information on country of birth, and undelivered invitation letters	Country of birth	Migrants and Italians who were residents in the Florence district invited to screening between 2000 and 2008
Webb et al., 2004 [47]	United Kingdom	Age: 25–49 years Time interval: 3 years Age: 50–64 Time interval: 5 years	Manchester Health Authority, and National Database of Primary Care Trusts	Women screened in the last 5 years	Absence of cervix	Country of birth	All eligible migrant and native women aged 30–64 years
Yeasmeen et al., 2019 [48]	Australia	Age: 25–74 years Time interval: 3 years	Victorian Cervical Cytology Registry, and Victorian Admitted Episodes Dataset	Women identified in both selected datasets was defined as a screening participant	Women who previously underwent hysterectomy or who died prior to the period of interest	Country of birth	Migrant and native women aged 15 years or over in the period from 1 January 2000 to 31 December 2013

**Table 3 ijerph-20-02200-t003:** Sample sizes for migrants and natives included in the studies.

Study	N Migrant Women	N Screened Migrant Women (%)	N Natives	N Screened Native Women (%)
Aminisani et al., 2012 [16]	11,477	6879 (59.9%)	10,762	6834 (63.5%)
Azerkan et al., 2012 [33]	445,547	220,247 (49.4%)	2,176,255	1,349,278 (62.0%)
Badre-Esfahani et al., 2020 [34]	18,273	4965 (27.2%)	151,885	78,903 (51.9%)
Battagello et al., 2022 [35]	27,958	10,443 (37.4%)	96,105	47,069 (49.0%)
Broberg et al., 2018 [36]	178,917	46,317 (25.9%)	369,574	218,036 (59.0%)
Comparetto et al., 2017 [37]	4992	555 (11.1%)	40,688	22,728 (55.9%)
Gallo et al., 2017 [38]	500,381	220,155 (44.0%)	4,017,764	1,955,373 (48.7%)
Harder et al., 2018 [39]	48,218	36,357 (75.4%)	428,452	366,627 (85.6%)
Hertzum-Larsen et al., 2019 [40]	57,329	34,844 (60.8%)	553,578	411,898 (74.4%)
Idehen et al., 2020 [41]	1790	926 (51.7%)	789	525 (66.5%)
Leinonen et al., 2017 [42]	208,626	106,399 (51.0%)	1,157,223	791,228 (68.4%)
Lofters et al., 2015 [25]	7737	5370 (69.4%)	31,268	22,482 (71.9%)
Pankakoski et al., 2020 [43]	129,049	78,658 (61.0%)	1,098,410	762,296 (69.4%)
Rodvall et al., 2005 [44]	67,581	38,071 (56.3%)	239,971	132,872 (55.4%)
Virtanen et al., 2015 [45]	1818	1283 (70.6%)	29,009	23,317 (80.4%)
Visioli et al., 2015 [46]	52,281	20,094 (38.4%)	488,498	228,129 (46.7%)
Webb et al., 2004 [47]	8921	5120 (57.4%)	15,937	12,048 (75.6%)
Yeasmeen et al., 2019 [48]	499,967	171,865 (34.4%)	1,298,494	489,532 (37.7%)

## Data Availability

Not applicable.

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
