# Peer review of "Adherence to Cervical Cancer Screening Programs in Migrant Populations: A Systematic Review and Meta-Analysis"

_ijerph, 2023, doi:10.3390/ijerph20032200_

Round 1
Reviewer 1 Report
Introduction
Line 34 – confusing sentence structure, unlikely of what?
Line 46 – disagree with the statement that screening with HPV versus cytology provides a longer protective effect. It may need to be clarified such as it decreases the frequency of screening and less chance for false positive and unnecessary subsequent testing. Sentence as is structured currently does not make clear how it is protective.
Materials and Methods
Please explain further why refugees and asylum seekers were not included in the analysis? Is there a certain cut off in number of years living in the host country to be considered similar in characteristics access to organized programs to be equivalent to migrant women?
There is definition included for “migrant” women although throughout the paper there is inconsistencies in which term is chosen, i.e. migrant, immigrant, foreign-born. Please update for consistent terminology.
Table 1
- can “others” be expanded in the exclusion criteria
- please explain why participation in self-sampling strategies was excluded
- inclusion criteria included only organized cervical cancer screening, were then opportunistic programs an exclusion criteria?
Line 117 – add s to paper > papers
Line 111 – states the research was done in two steps but describes three steps which is reflected in Figure 1 PRISMA flowchart
Results
Table 2
- fix consistencies between small or large capital characters
Line 163 – change main to mean
Discussion
Line 190 – recommend adding one or two more points to this paragraph
Line 194 – remove “ly” from commonly > common
Line 206 – remove “the” before income
Line 241 – sentence if confusing, what does categories refer to? Countries? Are you stating that because the population is less stable that the country doesn’t prioritize health services?
Line 242 – do you mean “This evidence..”
In the methods HMPCs and LMPCs were defined which I found to be interesting and expected this to be included in analysis either in the results or discussion regarding the significance or impact of coming from a HMPC vs LMPC.
Conclusion
Line 280 – recommend changing that to which
Author Response
Reviewer 1:
Introduction
- Line 34 – confusing sentence structure, unlikely of what?
The sentence was corrected and know reads as follows (page 1, line 33-34):
“Individuals are often unaware of the ongoing infection and precancerous lesions are mostly asymptomatic”.
- Line 46 – disagree with the statement that screening with HPV versus cytology provides a longer protective effect. It may need to be clarified such as it decreases the frequency of screening and less chance for false positive and unnecessary subsequent testing. Sentence as is structured currently does not make clear how it is protective.
The sentence was modified according to the reviewer’s suggestion and now reads as follows (page 1-2, lines 44-48):
“Screening performed using HPV-testing ensures a standard screening quality, while showing an increased sensitivity and accuracy compared to cervical cytology, with its high negative predictive value allowing a 5-year screening interval, longer than the 3-year interval required for Pap smear”.
Materials and Methods
- Please explain further why refugees and asylum seekers were not included in the analysis? Is there a certain cut off in number of years living in the host country to be considered similar in characteristics access to organized programs to be equivalent to migrant women?
We thank the reviewer for this comment. We added some explanations in the Materials and Methods section, regarding exclusion criteria, to further justify the decision of excluding studies on refugees and asylum seekers from the review (see page 3, lines 98-106):
“Studies that investigated only refugees and asylum seekers’ participation in cervical screening were excluded from the present review, because their characteristics differ greatly from those of other groups of migrants. Subjects migrating on the basis of socio-economic reasons make a conscious, voluntary choice to leave their country of origin and can eventually return home in safety, if desired. On the other hand, refugees and asylum seekers undergo forced migration, during their early resettlement they experiment unique challenges associated with health care utilization and access, and women forming part of these subgroups show different priority health issues compared to other migrant women”.
We further specify to the reviewer that studies involving asylum seekers and refugees’ participation in cervical screening were excluded when these subgroups were the only ones investigated.
- There is definition included for “migrant” women although throughout the paper there is inconsistencies in which term is chosen, i.e. migrant, immigrant, foreign-born. Please update for consistent terminology.
Thank you for the suggestion, the whole manuscript has been updated using the term “migrant” when necessary.
1.5 Table 1:
- can “others” be expanded in the exclusion criteria
- please explain why participation in self-sampling strategies was excluded
- inclusion criteria included only organized cervical cancer screening, were then opportunistic programs an exclusion criteria?
With respect to inclusion and exclusion criteria, we updated Table 1 as requested (see page 3), and provided explanations to the decision of excluding participation in self-sampling strategies (page 3, lines 111-116):
“We also excluded studies that reported participation in cervical screening through self-sampling strategies, because these are not included nowadays in all organized screening programs; moreover, in most of published studies, self-sampling approaches have been proposed as an alternative screening method to specific subgroups of non-attendees or women who actively decline traditional screening”.
We would like to also specify to the reviewers that we excluded studies that presented participation to cervical screening only through self-sampling strategies, but studies that presented information regarding both strategies (self-sampling and traditional approaches) were included, and only data regarding the second were extracted from the papers.
We also added participation in opportunistic screening among exclusion criteria, as suggested.
- Line 117 – add s to paper > papers
Line 111 – states the research was done in two steps but describes three steps which is reflected in Figure 1 PRISMA flowchart
We corrected the error and better described the steps forming part of the literature search (see page 3, lines 124-128):
“The research was conducted in three steps. In the first step, we identified and removed duplicates; in the second step, two reviewers (IR and VT) screened all identified papers independently by title and abstract, selecting only those that focused on cervical cancer screening among migrant women. The third step consisted in independently screening the selected papers by full text”.
Results
- Table 2: fix consistencies between small or large capital characters
Line 163 – change main to mean
Errors and inconsistencies were fixed according to the reviewer’s suggestion.
Discussion
- Line 190 – recommend adding one or two more points to this paragraph
The paragraph was modified according to the suggestion and now reads as follows (page 11, lines 214-217):
“Overall, the results confirm that migrant women, irrespective of their country of origin, have a significantly lower adherence to cervical screening compared with native women; screening participation is extremely low for women coming from North and Sub-Saharan Africa and higher for women coming from Central and South America”.
- Line 194 – remove “ly” from commonly > common
Line 206 – remove “the” before income
Line 241 – sentence if confusing, what does categories refer to? Countries? Are you stating that because the population is less stable that the country doesn’t prioritize health services?
Line 242 – do you mean “This evidence..”
We corrected the grammar errors and decided to eliminate the confusing sentence (page 12, lines 267-269).
- In the methods HMPCs and LMPCs were defined which I found to be interesting and expected this to be included in analysis either in the results or discussion regarding the significance or impact of coming from a HMPC vs LMPC.
We added a sentence to justify our decision to focus on HMPC countries only which reads as follow (page 12, lines 283-286):
“Migrants coming from low migratory pressure countries (LMPCs) were not considered for stratified analyses, because previously published studies show that they are generally characterized by a similar health profile to that of native populations”.
Conclusion
1.11 Line 280 – recommend changing that to which
We fixed the error.
Reviewer 2 Report
The authors have designed a review of cervical cancer screening compliance in migrant populations compared to native ones, complete with meta-analysis. I appreciate the effort provided in describing the selection process, as well as the care given to ensure the data was being compiled from as similar of studies as possible. The authors include much discussion regarding potential influences on comparative studies.
This was an extremely well-written manuscript, with appropriate data tables and statistical analyses. The authors have also clearly reviewed the language presentation, as the grammar throughout is impeccable.
I cannot find issue with any aspects of the manuscript, including the discussion. The authors provide several explanations to highlight why the migrant populations have lower screening rates, as well as suggestions to remedy these issues.
Well done!
Author Response
Reviewer 2:
The authors have designed a review of cervical cancer screening compliance in migrant populations compared to native ones, complete with meta-analysis. I appreciate the effort provided in describing the selection process, as well as the care given to ensure the data was being compiled from as similar of studies as possible. The authors include much discussion regarding potential influences on comparative studies.
This was an extremely well-written manuscript, with appropriate data tables and statistical analyses. The authors have also clearly reviewed the language presentation, as the grammar throughout is impeccable.
I cannot find issue with any aspects of the manuscript, including the discussion. The authors provide several explanations to highlight why the migrant populations have lower screening rates, as well as suggestions to remedy these issues.
Well done!
We thank Reviewer 2 for the overall positive evaluation of the manuscript.
Reviewer 3 Report
- Introduction: This section is well written, comprehensive and informative. The knowledge gap is clearly outlined. The importance of the topic is explained.
- Line 5: What is meant by "migrant women with suitable characteristics"?
- Lines 89-90: The FigureS1 could easily be a table instead, this is a picture of the search string.
- Line 90: It is not enough to provide a search string for one database only. Add search strategies for all three databases that were searched.
- Table 1: For inclusion criteria, check whether it is correct to state that there were no restrictions on publication year - rather be specific and state that papers that were published up to uly 2022 when you last searched the databases were considered.
- Lines 127-130: Explain whether you pooled adjusted or unadjusted ORs.
- Line 133: Provide the cut-off values for I2 statistic depicting what you considered as different levels of heterogeneity. Add an appropriate reference.
- Lines 129-130: Clarify - were these not readily available in any of the included studies?
- Methods: Explain how publication bias was assessed.
- Methods: Explain whether and how the results were presented graphically for the ran meta-analyses.
- Figure 1: Could you clarify what the reason for exclusion "not available text" means? Not available for free or?
- Table S1: Specify for the last column whether these are adjusted or unadjusted ORs. If they are adjusted, specify for which variables they were adjusted.
- Methods: Heterogeneity is very high, 100%. Warrants further exploration using meta-regression techniques.
- Since the paper followed the guidelines, add the PRISMA checklist as a supplementary file.
- Limitations: Including only papers in English language is a limitation too.
Author Response
Reviewer 3:
- Introduction: This section is well written, comprehensive and informative. The knowledge gap is clearly outlined. The importance of the topic is explained.
Line 5: What is meant by "migrant women with suitable characteristics"?
We thank the reviewer for the comment.
Line 55 was corrected (see now page 2, line 87): “women with suitable characteristics” is corrected into “women with the correct age range for participation”.
- Lines 89-90: The FigureS1 could easily be a table instead, this is a picture of the search string.
Line 90: It is not enough to provide a search string for one database only. Add search strategies for all three databases that were searched.
As suggested by the reviewer, information regarding the search string was provided in Table S1 for all databases searched (Supplementary).
- Table 1: For inclusion criteria, check whether it is correct to state that there were no restrictions on publication year - rather be specific and state that papers that were published up to July 2022 when you last searched the databases were considered.
We corrected the sentence in Table 1, stating that we included papers published up to July 2022 (see page 3).
- Lines 127-130: Explain whether you pooled adjusted or unadjusted ORs.
Line 133: Provide the cut-off values for I2 statistic depicting what you considered as different levels of heterogeneity. Add an appropriate reference.
We added the requested information to the manuscript together with the appropriate reference (see page 4, lines 146-150):
“To account for the variability in settings and populations investigated in the included studies, we chose a random-effects model for the meta-analysis and pooled unadjusted ORs for screening participation in a forest plot. Heterogeneity among studies was investigated using I² statistic and was defined as low, moderate or high using I² cut-offs of 25%, 50% and 75%, respectively”.
- Lines 129-130: Clarify - were these not readily available in any of the included studies?
The requested clarification was added to the manuscript and the sentence now reads as follows (page 4, lines 143-145):
“When ORs were not readily available, we calculated unadjusted ORs (with relative standard error) for cervical screening participation among migrant women, compared to native women”.
- Methods: Explain how publication bias was assessed.
The presence of publication bias was assessed with a funnel plot, that has been added to Supplementary Material (see Figure S3).
To check for the presence of publication bias, we visually verified the absence of funnel plot asymmetry. We also used the regression test to check of funnel plot asymmetry, that reported a not statistically significant p-value of 0.2873.
Information regarding publication bias assessment was included in the manuscript in the Materials and Results sections (see page 4, lines 161-162 and page 10, lines 207-208):
“Publication bias was evaluated checking for the presence of asymmetry in the funnel plot and using the regression test”.
“Funnel plot examination and regression tests did not show asymmetry (Figure S3)”.
- Methods: Explain whether and how the results were presented graphically for the ran meta-analyses.
We added the requested information (see page 4, lines 147-148):
“we chose a random-effects model for the meta-analysis and pooled unadjusted ORs for screening participation in a forest plot”.
We further specified that forest plots were provided for stratified analyses (see page 4, line 150).
- Figure 1: Could you clarify what the reason for exclusion "not available text" means? Not available for free or?
With respect to Figure 1, papers marked as excluded because “not available” were excluded because the full text was not available for free for consultation, as the reviewer suggested.
- Table S1: Specify for the last column whether these are adjusted or unadjusted ORs. If they are adjusted, specify for which variables they were adjusted.
In the Table that now is named Table S2 (see Supplementary), ORs reported in the last column are reference ORs for both unadjusted and adjusted models, we reported this information in the heading.
- Methods: Heterogeneity is very high, 100%. Warrants further exploration using meta-regression techniques.
As heterogeneity is very high and stratification according to macro-areas of origin did not reduce the observed heterogeneity among studies, we conducted various meta-regression models using the variables extracted from the included studies, as suggested by the reviewer.
Univariate meta-regression models conducted considering 1) the geographic area in which the studies were conducted (North Europe vs Other), 2) the period of time regarding which data were collected for the study (number of years considered for data extraction from administrative sources in every study), 3) mean age of participants in years, 4) type of test used (Pap test vs Pap test and HPV-DNA test), 5) definition of adherent subject, 6) definition of migrant subject and 7) publication year, did not reduce the observed heterogeneity.
We argue that the observed heterogeneity could be associated with other aspects, such as the different composition of migrant groups involved in every study, together with the differences in migrants’ classification methods, that have been presented in Table S2. This information cannot be included in meta-regression models as it is too various to be categorized adequately.
Variable |
|
Estimate |
p-value |
95%CI |
Tau² |
I² |
Country (ref: North Europe vs Other) |
|
0.08 |
0.76 |
(-0.45; 0.61) |
0.32 |
99.98% |
Period of time considered (years) |
|
-0.01 |
0.89 |
(-0.08;0.07) |
0.32 |
99.98% |
Mean age (years) |
|
-0.003 |
0.86 |
(-0.05;0.04) |
0.37 |
99.98% |
Type of test used (ref: Pap test only vs Pap test and HPV-DNA) |
|
0.21 |
0.61 |
(-0.62; 1.04) |
0.32 |
99.99% |
Definition of adherent subject (ref: Tested in 1 year) |
Tested in 3 years Tested in 4-5 years Tested post invitation |
-0.16 -0.36 -0.82 |
0.66 0.33 0.04 |
(-0.87; 0.56) (-1.11; 1.38) (-1.59; -0.04) |
0.27 |
99.98% |
Definition of migrant subject (ref: country of origin/birth) |
Citizenship Mother tongue More than one criteria |
-1.76 0.09 0.22 |
<0.0001 0.74 0.45 |
(-2.55; -0.96) (-0.48; 0.67) (-0.36; 0.79) |
0.38 |
99.96% |
Publication year |
|
-0.02 |
0.41 |
(-0.07; 0.03) |
0.31 |
99.98% |
Information regarding the meta-regression models was included in the manuscript in the Materials and Methods section (page 4, lines 160-161):
“Meta-regression models and stratified analysis were conducted in the attempt to address the heterogeneity of the included studies”.
We also added the following sentence in the Results section (page 10, lines 203-207):
“Meta-regression models conducted using participants’ mean age, the definition of migrant subjects (country of birth/origin, citizenship, mother tongue or more than one of the previous), the type of test used (Pap test only or Pap test and HPV-DNA test) or publication year as covariates did not show a significant reduction in the observed heterogeneity”.
- Since the paper followed the guidelines, add the PRISMA checklist as a supplementary file.
As requested, we added the PRISMA checklist in Supplementary Materials (see TableS3).
- Limitations: Including only papers in English language is a limitation too.
In the Discussion section (page 12, lines 299-301), we included among the limitations the decision of including only papers written in English:
“The inclusion of papers written only in English might have led to miss information regarding participation to screening in migrants reported in other languages”.
These papers were excluded in the full-text screening phases and there were 5 of them in total.